# A Novel *Drosophila* Model of Alzheimer’s Disease to Study Aβ Proteotoxicity in the Digestive Tract

**DOI:** 10.3390/ijms25042105

**Published:** 2024-02-09

**Authors:** Greta Elovsson, Therése Klingstedt, Mikaela Brown, K. Peter R. Nilsson, Ann-Christin Brorsson

**Affiliations:** Department of Physics, Chemistry, and Biology, Linköping University, 581 83 Linköping, Sweden; greta.elovsson@liu.se (G.E.); therese.klingstedt@liu.se (T.K.); mikaela.e.brown@gmail.com (M.B.); peter.r.nilsson@liu.se (K.P.R.N.)

**Keywords:** Alzheimer’s disease, *Drosophila*, amyloid-beta (Aβ)

## Abstract

Amyloid-β (Aβ) proteotoxicity is associated with Alzheimer’s disease (AD) and is caused by protein aggregation, resulting in neuronal damage in the brain. In the search for novel treatments, *Drosophila melanogaster* has been extensively used to screen for anti-Aβ proteotoxic agents in studies where toxic Aβ peptides are expressed in the fly brain. Since drug molecules often are administered orally there is a risk that they fail to reach the brain, due to their inability to cross the brain barrier. To circumvent this problem, we have designed a novel *Drosophila* model that expresses the Aβ peptides in the digestive tract. In addition, a built-in apoptotic sensor provides a fluorescent signal from the green fluorescent protein as a response to caspase activity. We found that expressing different variants of Aβ1–42 resulted in proteotoxic phenotypes such as reduced longevity, aggregate deposition, and the presence of apoptotic cells. Taken together, this gut-based Aβ-expressing fly model can be used to study the mechanisms behind Aβ proteotoxicity and to identify different substances that can modify Aβ proteotoxicity.

## 1. Introduction

Alzheimer’s disease (AD) is a widespread, neurological disorder that involves an extensive neuronal loss in the brain followed by cortical and hippocampal atrophy [1,2]. The two main pathological hallmarks of AD are neurofibrillary tangles and amyloid plaques, where the latter is essentially composed of the amyloid-β (Aβ) peptide [3]. Aβ easily misfolds and aggregates into neurotoxic oligomers, which merge into insoluble amyloid fibrils [4]. The amyloid cascade hypothesis postulates that the major cause of AD is accelerated production and deposition of Aβ [5,6,7]. Aβ originates from the amyloid-β precursor protein (AβPP) and is generated through sequential cleavages by β-site AβPP-cleaving enzyme (BACE1) and γ-secretase [8,9]. Depending on the γ-secretase cleavage site, different Aβ isoforms are produced, where Aβ1–40 and Aβ1–42 are most common [10]. Aβ1–42 plays a crucial role in AD due to its hydrophobic and amyloidogenic nature [7,11]. The Arctic mutation (Glu22Gly) of the Aβ peptide is associated with higher neurotoxicity and an accelerated aggregation rate of Aβ, manifesting in a severe form of AD [12]. In vitro studies of various Aβ variants have increased our understanding of the aggregation mechanisms of the Aβ peptide [4,13,14]. Interestingly, the dimeric form of Aβ has been suggested to be involved in the initial part of the aggregation process [15]. In a study from 2012, various tandem constructs of two Aβ peptides, linked together to simulate the Aβ dimer, were investigated for their proteotoxic effect in *Drosophila* [16]. Tandem constructs of Aβ1–42 showed increased in vivo toxicity and higher level of insoluble aggregates compared to monomeric Aβ1–42. The tandem construct with the most progressive and toxic traits was the one with a 22 amino acid long linker in between two Aβ1–42 peptides (T_22_Aβ1–42).

Despite millions of people being afflicted with AD worldwide and extensive research in this field, there is still no cure for AD. The use of the fruit fly *Drosophila melanogaster* to model AD, and other neurodegenerative diseases, has been widely applied to study the pathological mechanisms and to screen for disease-modifying agents in the search for therapeutic approaches [17,18]. The Gal4/UAS system makes it possible to express a target protein in a specific tissue or cell type and a variety of phenotypes, which are associated with proteotoxicity similar to the disease progression in humans, can be studied in the flies [18,19]. In traditional drug screens against Aβ toxicity using *Drosophila*, the potential therapeutic compound is mixed in the food and administered to flies expressing the Aβ peptide in the central nervous system (CNS) by exploiting the embryonic lethal abnormal vision (*elav*)-Gal4 driver [20,21,22]. Then, the flies are examined to verify whether the drug can rescue phenotypes that are associated with the presence of toxic Aβ species in the fly brain such as reduced lifetime [21,23,24], decreased locomotor function [21,23,24], increased/decreased protein levels [24,25], oxidative stress [25], apoptosis [25,26], and accumulation of protein aggregates [24,27,28]. In these screens, however, there is great uncertainty as to whether the drug can cross the fly’s equivalency to the human blood-brain barrier and reach the fly brain to exert its antitoxic effect. Thus, the lack of a detectable rescue effect of a drug may not be due to the inability of the drug to block Aβ toxicity but rather that the drug cannot enter the brain area of the flies where the toxic events take place. To overcome this problem, we have developed a novel *Drosophila* model of AD where the Aβ peptide is expressed in the fly intestine. In this way, an encounter between the orally administered drug and the toxic protein will be more probable to enable a correct investigation of whether the tested compound can modulate the toxic properties of the protein. In this AD fly model, the fly driver line *Myo31DF* is used, which expresses the target protein in the enterocytes in the fly’s digestive tract [29,30,31]. Additionally, the apoptotic sensor *UAS-GC3Ai* is integrated into the Aβ fly genotype, which provides a fluorescence signal from the green fluorescent protein (GFP) as a response to caspase activity [26]. Caspase activation is one of the main downstream effects that occurs due to apoptotic cell death [32].

In the present study, three different Aβ fly genotypes were constructed that expressed either two copies of the Aβ1–42 peptide (henceforth referred to as Aβ1–42 × 2 flies), one copy of the tandem Aβ1–42 construct with a 22 amino acid long linker (henceforth referred to as T_22_Aβ1–42 flies) or one copy of the Aβ1–42 Arctic mutant (henceforth referred to as Arctic flies) in the fly gut using the *Myo31DF* driver (with the *UAS-GC3Ai* construct). Additionally, w1118 flies were crossed with the *Myo31DF* driver to create control flies. Proteotoxic effects by expression of the three Aβ peptides in the fly gut were examined by a longevity assay, GFP fluorescence to identify apoptotic cells, and by specific staining of Aβ aggregates using an antibody and the amyloid-binding luminescent conjugated oligothiophene (LCO) h-FTAA [33,34]. Toxic effects were found for all three Aβ peptides, which manifested in reduced lifespan, the presence of apoptotic cells, and the formation of Aβ aggregates. The longevity assay revealed a higher toxic effect for the Arctic- and T_22_Aβ1–42 flies compared to the Aβ1–42 × 2 flies, which is in line with previous research [16,21]. The highest amount of aggregates was detected for the T_22_Aβ1–42 flies, followed by Arctic flies, and lastly by the Aβ1–42 × 2 flies. In conclusion, we have developed a new *Drosophila* model of AD, based on expression of Aβ in the fly intestine, which exhibits similar toxic effects to previous neuronal Aβ-expressing fly models and can be used to advance our understanding of the mechanism of Aβ proteotoxicity. Furthermore, this gut-based AD fly model increases the likelihood that orally administered substances reach the target site, where the Aβ toxicity originates, in the search for compounds that can modulate Aβ proteotoxicity.

## 2. Results

### 2.1. Shortened Lifespan in Aβ-Expressing Flies

A common method to investigate proteotoxicity in *Drosophila* is to perform a survival assay where the median survival time (day when 50% of all flies are dead) is compared between flies expressing the proteotoxic protein and control flies [18,23,35]. Therefore, to investigate the proteotoxic effect of expressing Aβ in the enterocytes of the fly’s digestive tract, a lifetime analysis was performed. Data from this experiment showed that all Aβ-expressing flies had a significantly reduced longevity compared to the control flies (Figure 1); Arctic-, T_22_Aβ1–42-, and Aβ1–42 × 2 flies received a reduction in median survival time of 12 (*p* < 0.0001), 11 (*p* < 0.0001), and 6 (*p* < 0.0001) days, respectively. The median survival times for Arctic-, T_22_Aβ1–42-, Aβ1–42 × 2-, and control flies were 15, 16, 21, and 27 days, respectively, revealing that expression of Aβ in the fly gut resulted in toxic effects that reduced the lifetime of the fly. Moreover, a significant difference in the longevity between Aβ1–42 × 2 flies and the other two Aβ-expressing flies was found, where the Arctic- and T_22_Aβ1–42 flies exhibited higher toxicity, resulting in a reduction of 6 days (Arctic flies) or 5 days (T_22_Aβ1–42 flies) in median survival time compared to Aβ1–42 × 2 flies. These data are in line with previous research, where a higher toxic effect has been found for the Arctic Aβ1–42 peptide and the tandem construct T_22_Aβ1–42 compared to wildtype Aβ1–42 peptide [16,18,21,27].

### 2.2. Detection of Aggregates in Aβ-Expressing Flies

Next, we wanted to investigate the presence of Aβ aggregates in the fly gut and examine how they relate to toxicity. To achieve this, co-staining was performed using an antibody against Aβ and the amyloid-binding LCO ligand h-FTAA [33,34]. The experiment was carried out at time points that corresponded to the median survival time for the different fly genotypes (day 15 for the Arctic flies; day 16 for the T_22_Aβ1–42 flies; day 21 for the Aβ1–42 × 2 flies). If not specified, all images were acquired from the anterior midgut as displayed in Figure 2. In all Aβ-expressing flies, Aβ aggregates were detected by the antibody in the midgut of the flies and these aggregates also gave a positive h-FTAA signal, indicating the presence of a cross-β-sheet structure (Figure 3A, Figure 4A and Figure 5A). Since the Aβ aggregates were located at different depths in the fly gut, a 3D image (z-stack) of the stained tissue was generated to provide a more accurate image of the aggregates, regarding both the amount and their morphology (Figure 3B, Figure 4B and Figure 5B).

Interestingly, the amount of detected Aβ aggregates varied substantially between the three different Aβ genotypes: T_22_Aβ1–42 flies had the greatest amount, followed by Arctic flies, and lastly Aβ1–42 × 2 flies where very few aggregates were detected. The anterior midgut of the T_22_Aβ1–42 flies was almost completely full of Aβ aggregates. In the Arctic flies, several aggregates, with different sizes, were found while only a few aggregates, which were quite small, were detected in the Aβ1–42 × 2 flies. From visual observation, the Aβ aggregates seem to appear intracellularly as they are in close vicinity to the cell nuclei. No signals that could be attributed to Aβ aggregates were found in the control flies. Diffuse signals from the Aβ antibody were observed in the midgut of the control flies; although, this is most likely due to unspecific binding of the antibody since no signal was detected from h-FTAA in this region. However, posterior to this region, structures showing h-FTAA fluorescence could be seen in all Aβ-expressing flies and in the control flies (Appendix A). These h-FTAA signals could be due to precipitation of the ligand in this area. In contrast to the mammalian digestive system, *Drosophila melanogaster* has generally a neutral or mildly alkaline pH in the intestine; however, in a smaller region in the middle of the midgut, where copper cells are located, the pH is strongly acidic [36]. This acidic region is situated approximately in the same area as the undefined h-FTAA signals. Therefore, the fluorescence signals in this area are probably a result of precipitation of h-FTAA, due to the acidic milieu, rather than evidence of Aβ aggregates since no signal from the Aβ antibody was detected in this area.

### 2.3. Increased Number of Apoptotic Cells in Aβ-Expressing Flies

To further study the toxicity of these flies, the built-in *UAS-GC3Ai* apoptotic sensor was exploited [26]. This sensor makes it possible to monitor the presence of apoptotic cells in fly tissue, due to a fluorescence signal of GFP as a response to caspase activity. The expressed protein GC3Ai resembles native GFP with one exception: GC3Ai is non-fluorescent due to joining of the C- and N-terminus by a sequence containing a DEVD caspase cleavage site. Consequently, caspase cleavage is necessary for GC3Ai to regain a fluorescent property that is identical to GFP [26]. The analysis of apoptotic cells in the Aβ-expressing flies was performed at day 15 for the Arctic- and T_22_Aβ1–42 flies, and at day 21 for the Aβ1–42 × 2 flies. Apoptotic cells were found in the midgut area in the same region where the Aβ aggregates were detected in all Aβ fly variants (Figure 6). Apoptotic cells were also detected in the control flies (Figure 6). While the GFP fluorescence of the Arctic- and T_22_Aβ1–42 flies only appeared to be slightly more abundant compared to their respective control flies at day 15, the number of GFP-positive cells in the Aβ1–42 × 2 flies was substantially higher compared to the control flies at day 21. The minor differences between the controls on day 15 and day 21 could be explained by the diversity between experiments.

When comparing these three Aβ-expressing flies, the number of GFP-positive cells was higher in the Aβ1–42 × 2 flies compared to the other two Aβ genotypes. Additionally, flies without the driver gene *Myo31DF* and the apoptotic sensor were studied to gain a GFP expression baseline (Appendix A). In these flies, no fluorescence signals were detected.

## 3. Discussion

*Drosophila melanogaster* is an in vivo model organism extensively used to investigate pathological mechanisms underlying neurodegenerative diseases, e.g., AD, and to find substances that can counteract the toxicity derived from these diseases [17,18]. Organs in the fly, such as brain and gut, and the functionality of the neuronal network are similar to their respective counterparts in mammals [37]. The Gal4/UAS system makes it possible to direct gene expression to a specific cell type or tissue in the fly, which is highly useful when studying proteotoxicity [19]. Indeed, a wide range of phenotypic markers related to proteotoxicity, such as decreased longevity and the presence of aggregates, are available in *Drosophila* and enable studies of toxic mechanisms. Other advantages when using a *Drosophila* model are a relatively short lifespan, low maintenance, and a high number of individuals, which gives good statistics. In most AD *Drosophila* models, the Aβ peptide expression is directed to the CNS of the flies resulting in reduced longevity, Aβ aggregate accumulation, and neuronal death [23,25]. Using this model, a correlation has been discovered between the degree of the proteotoxic effect of different Aβ peptides in *Drosophila* and their aggregation properties [21,27,28]. One of the most toxic Aβ peptides is the Arctic mutation (Glu22Gly) of the Aβ1–42 peptide. The Glu22Gly amino acid substitution causes a more rapid formation of neurotoxic protofibrils and delays the formation of mature amyloid fibrils, which are believed to be inert towards the cells [12,38]. In a study from 2012, gene constructs of tandem Aβ (two Aβ peptides linked together to mimic dimeric species of Aβ) were expressed neuronally in *Drosophila* to investigate their neurotoxic properties [16]. Two of these tandem constructs, composed of Aβ1–42 peptides, were found to accelerate the aggregation process leading to a greatly reduced lifespan of the flies.

Results from our study show that there are many similarities between our gut-based AD model and the traditional neuronal-based AD model regarding proteotoxic effects of Aβ. The lifespan analysis clearly showed that Aβ has a toxic effect on the flies as the median survival time was reduced by 12 days for the Arctic flies, 11 days for the T_22_Aβ1–42 flies, and 6 days for the Aβ1–42 × 2 flies compared to control flies. These results resemble the reduction in the lifespan of Aβ-expressing flies when using the neuronal fly driver *elav-*Gal4, where flies expressing the Arctic mutation of Aβ1–42 or T_22_Aβ1–42 in the CNS have significantly lower median survival time compared to wildtype Aβ1–42-expressing flies [16,21,27,35].

It is well-known that there is a strong correlation between Aβ proteotoxicity and Aβ aggregation [16,18,21,27,28,39,40]. Pro-aggregatory Aβ peptides are responsible for toxic effects such as reduced lifespan, locomotor deficits, and destroyed tissue caused by neurodegeneration in *Drosophila*. Crowther and his team found that expression of Arctic Aβ1–42 or two copies of Aβ1–42 in flies leads to premature death and severe eye phenotypes, as a result of Aβ accumulation and formation of aggregates [21]. They found that flies expressing Aβ1–40 exhibited significantly less proteotoxicity compared to flies expressing Aβ1–42, due to Aβ1–40 being less prone to aggregation. In a study from 2015, various Aβ peptides were tested in *Drosophila*, and the results showed that the Aβ aggregation load was in agreement with the toxicity [28]. By studying truncated, extended, and mutated Aβ peptides, they were able to underscore the importance of amino acid Ala42 in regard to Aβ aggregation. Histology analyses in our study revealed that expression of all three Aβ1–42 variants under the control of *Myo31DF* resulted in Aβ aggregates, which were detected by an anti-human Aβ antibody and the LCO ligand h-FTAA. As expected, no structures identifiable as Aβ aggregates were visualized in the control flies. These data are in line with previous studies where Aβ expression in the CNS resulted in the formation of Aβ aggregates [11,16,21,27]. In our study, a greater amount of aggregates was found in the Arctic flies compared to the Aβ1–42 × 2 flies; Arctic flies showed an accumulation of Aβ aggregates in several regions of the intestine while the intestine of Aβ1–42 × 2 flies was often lacking aggregates or contained only a small amount. This is consistent with previous results from histological experiments using a neuronal-based AD model, where the Arctic mutation resulted in an accelerated aggregation rate and increased accumulation of aggregates compared to the Aβ1–42 [21]. The greatest aggregation load was found in the T_22_Aβ1–42 flies, where the intestine was almost completely full of aggregates in contrast to both Arctic- and Aβ1–42 × 2 flies which contained a lower amount. This result is consistent with previous research where an increased amount of Aβ aggregates was found for T_22_Aβ1–42 compared to monomeric Aβ1–42 when expressed in the neurons using the *elav*fly driver line [16]. The fact that T_22_Aβ1–42 flies had a similar median survival time as the Arctic flies, albeit a significantly higher number of Aβ aggregates, suggests that in this case, the aggregation load does not correlate with the level of toxicity. This is an interesting phenomenon indicating that there are aggregates in the T_22_Aβ1–42 flies with lower toxicity than in Arctic flies, revealing that Aβ aggregates produced in *Drosophila* may differ in their toxicity. A similar result was found in a previous study where Luheshi and her team expressed a variety of Aβ peptides in the fly brain and mapped the correlation between amyloid deposition and toxicity for each Aβ variant [41]. They found that the mutation I31E on the Arctic (E22G) Aβ1–42 peptide resulted in significantly prolonged survival compared to Arctic (E22G) Aβ1–42, despite having a similar aggregation load, revealing that Aβ aggregates produced by the I31E mutated variant of the Arctic (E22G) Aβ1–42 peptide possesses less toxicity than aggregates formed by the Arctic (E22G) Aβ1–42 peptide. Similarly, Speretta et al. discovered that producing a tandem construct of Aβ1–40 in the fly brain increased aggregation but did not cause any toxic effect [16]. There is also one study on curcumin’s ability to mitigate the pathological effects caused by Aβ expression in *Drosophila* [24]. They found that curcumin reduced the neurotoxicity even though Aβ fibrillation was promoted. Taken together, this suggests that the level of toxicity depends on which kinds of aggregates are being produced rather than the aggregation load per se. Therefore, to find a therapeutic drug against AD, it is important to specifically target those Aβ aggregates that are responsible for toxicity.

The fly driver line used in this study (*Myo31DF*) has another feature besides allowing for the expression of the target protein in the fly’s intestine. It also contains a sensor construct that gives a fluorescent signal from a GFP-based protein upon caspase activity, making it possible to study the presence of apoptotic cells, which is a good phenotypic marker for neurodegeneration and proteotoxicity [25,26]. Fluorescence analyses using this sensor showed that the Aβ1–42 × 2 flies had a larger amount of apoptotic cells in the midgut compared to control flies at the same age, which barely showed any signs of apoptosis. This result shows that expression of Aβ1–42 is toxic to the cells which is in line with a previous study where apoptotic cells were detected when expressing Aβ1–42 in the neurons [25]. Theoretically, higher toxicity should result in more apoptotic cells. However, it is important to keep in mind that other events, e.g., necrosis, can play a role in premature cell death. In contrast to apoptosis, which is a form of programmed cell death, necrosis is defined as uncontrollable cell death in response to sudden damage [42,43]. It is difficult to anticipate the preferred cell death pathway due to Aβ proteotoxicity since apoptosis and necrosis are closely related events and thereby often co-exist. For apoptosis to occur, ATP must be available. Therefore, if the ATP supply is exhausted somewhere along the apoptotic process, secondary necrosis will take place instead, thus forcing the cell to undergo lysis [43]. Surprisingly, the Arctic- and T_22_Aβ1–42 flies showed almost as small amounts of apoptotic cells in the intestine as the control flies at the same time point despite showing a high toxicity in the survival assay. The reason for this might be that necrosis was favored over apoptosis due to the immediate toxicity emerging in these two apparently toxic fly genotypes. If necrosis took place instead of apoptosis, the GFP fluorescent signal would consequently be mute. On the contrary, the toxicity of the Aβ1–42 × 2 flies appears to be less acute since the Aβ1–42 × 2 flies lived longer and had fewer aggregates compared to the Arctic- and T_22_Aβ1–42 flies, which might suggest that apoptosis was favored over necrosis in the Aβ1–42 × 2 flies. Further studies are required to confirm the preferred cell death pathway that takes place in the Aβ-expressing flies used in this study. Another theory that could explain the remarkably few GFP-positive cells in the Arctic- and T_22_Aβ1–42 flies is that the produced Aβ aggregates interact with GFP causing conformational changes that abolish the GFP fluorescence. The control flies showed signs of apoptosis which was expected since it is a natural process in the midgut epithelium tissue [44]. We also found that the amount of apoptotic cells was reduced in the control flies over time. This could be explained by an inferior renewal of enterocytes in older tissue [45]. Therefore, the window of observing apoptotic cells is broader in younger tissue where cells are being more continuously replenished.

Expression of Aβ in the neurons of *Drosophila* is often used to perform drug screens to find anti-Aβ proteotoxic candidates [18]. In a study from 2005, the dye Congo Red showed a rescue effect in flies expressing the Aβ1–42 peptide or the Arctic mutant of the Aβ1–42 peptide in neuronal tissue. Histological analyses revealed that Congo Red alleviated Aβ toxicity by reducing the number of aggregates in the Aβ-expressing flies [21]. Inhibiting the aggregation process could indeed be an effective therapeutic strategy, since oligomeric species are highly associated with proteotoxicity [39]. The opposite approach, yet with a similar outcome, is to promote the conversion of toxic oligomeric species into inert amyloid fibrils, thus mitigating Aβ toxicity. Curcumin has been shown to have this effect on Aβ-expressing flies [24]. Another therapeutic strategy for AD is to increase protein clearance and degradation of Aβ, which was found to be the antitoxic mechanism for an engineered Aβbinding affibody protein [46]. They revealed that a specific construct of the affibody protein, two copies connected head-to-tail, almost completely abolished the neurotoxic effects when co-expressed with Aβ in the fly brain.

When using a *Drosophila* model to test a protein as a potential drug candidate, one main advantage is that the drug-protein and the proteotoxic protein can be co-expressed, providing a simultaneous occurrence in the fly tissue. Similar to the affibody, the protein lysozyme was tested for its anti-proteotoxic effects in such a way [35,47]. Lysozyme showed a rescue effect on neuronal-based AD fly models when studying rough eye phenotypes, longevity, and locomotor activity. In addition, Aβ levels were reduced in the presence of lysozyme, indicating that the anti-proteotoxic effects of lysozyme are due to its interaction with Aβ, and thus hindering the aggregation process. For non-protein compounds, oral administration is currently the most practical option to investigate their anti-toxic effect in Aβ-expressing flies. However, since a neuronal-based AD model produces the Aβ peptides in the fly brain, there is a risk that compounds, that are administered orally, will not encounter the toxic Aβ species due their inability to pass the blood-brain barrier in *Drosophila melanogaster* (DmBBB). This imminent risk makes negative results from a non-protein drug screen using a neuronal-based AD model difficult to interpret; if the substance does not cross the DmBBB, it would lead to false negative results. Thus, there is a possibility that potential anti-Aβ proteotoxic candidates will be missed on the screen. There appears to be no clear pattern of which properties a substance must have to be favored in transport across the DmBBB [48]. One possibility might be to attach lipids to the therapeutic compounds to mediate passage to the CNS [49]. DmBBB is constituted by layers of glial cells [50,51]. Naturally, its main purpose is to regulate transport of ions and nutrients, and to exclude, e.g., xenobiotics from the CNS. Interestingly, the permeability of DmBBB seems to be affected by a daily cycle; during nighttime the efflux from CNS is suppressed, thus retaining xenobiotics in the brain at a greater extent than during the daytime when the efflux is increased [52]. The uncertainty of whether a substance crosses the DmBBB complicates the experiment if the selected target site is the fly brain. This problem is circumvented when using our gut-based *Drosophila* model of Aβ toxicity where the toxic Aβ species is present in the digestive tract which increases the possibility that the orally administered substance and Aβ will interact.

Taken together, we have examined the proteotoxic effect of expressing Aβ1–42, the Arctic mutant (Glu22Gly) of the Aβ1–42 peptide and the tandem construct T_22_Aβ1–42 in the intestine of *Drosophila* using the driver line *Myo31DF*. There is a close relationship between longevity and production of Aβ aggregates, where greater amounts of Aβ aggregates in the intestine correlate with lower survival. We found that the Arctic- and T_22_Aβ1–42 flies had a higher accumulation of Aβ aggregates and lower median survival time compared to Aβ1–42 × 2 flies. However, despite a very similar medium survival time between Arctic- and T_22_Aβ1–42 flies (15 and 16 days, respectively) the aggregate load in the T_22_Aβ1–42 flies was substantially higher than in the Arctic flies revealing that the level of toxicity depends on which kinds of aggregates are being produced rather than the aggregation load per se. The amount of apoptotic cells did not correlate with a higher toxicity since the highest amount of apoptotic cells was detected in the Aβ1–42 × 2 flies which had a higher median survival compared to the Arctic- and T_22_Aβ1–42 flies. As previously discussed, the rather low GFP fluorescence seen in the Arctic- and T_22_Aβ1–42 flies, despite the high proteotoxic effect detected for these flies in the longevity assay, could be due to the fact that cell death occurs by necrosis instead of apoptosis. Another possible explanation is that an unexpected interaction occurs between the fluorescence protein and Aβ aggregates that abolishes the fluorescence signal in these flies.

In conclusion, we have developed a novel *Drosophila* model of AD, based on the expression of Aβ in the fly gut, that exhibits similar toxic effects to previous neuronal-expressing Aβ fly models. In addition, we found that different Aβ aggregates vary in their toxic properties. This gut-based Aβ-expressing fly model has a high potential to be used to advance our understanding of the formation of toxic Aβ species and to screen for compounds against Aβ proteotoxicity, since the expression of Aβ in the digestive tract increases the possibility that the drugs or substances interact with Aβ.

## 4. Materials and Methods

### 4.1. Drosophila Stocks

To achieve tissue- and cell-specific expression in *Drosophila melanogaster*, the Gal4/UAS system was used together with the driver line, *Myo31DF* (with GFP apoptotic sensor; Bloomington: 84307). This allows for protein expression in the enterocytes in the digestive tract of the fly and GFP signal during caspase activation. Fly lines carrying a double copy of signal peptide Aβ1–42 (Aβ1–42 × 2 flies), the tandem dimeric construct T_22_Aβ1–42 (T_22_Aβ1–42 flies) or the Arctic mutant (Glu22Gly) of the Aβ1–42 peptide (Arctic flies) were kindly provided by D. Crowther (AstraZeneca, Floceleris, Oxbridge Solutions Ltd., London, United Kingdom) and generated as described [16,21,25]. Moreover, w1118 flies (only expressing Gal4) were used as a control for Aβ1–42 × 2 flies, T_22_Aβ1–42 flies, and Arctic flies. The driver line *Myo31DF*, with a GFP apoptotic sensor, and control w1118 flies were purchased at Bloomington Stock Center. Fly crosses were set up at 25 °C at 60% humidity with 12:12-h light:dark cycles. Upon eclosion, flies were selected and reared in 29 °C at 60% humidity with 12:12-h light:dark cycles, using female offspring for staining and apoptotic assay and male offspring for longevity assay. For immunohistochemistry, h-FTAA staining, and GFP detection, flies were aged for 15 and 16 days (Arctic flies), 15 days (T_22_Aβ1–42 flies) or 21 and 24 days (Aβ1–42 × 2 flies), to match their respective median survival times, on regular fly food of corn meal, yeast, molasses, and agar.

### 4.2. Longevity Assay

Sets of 110–150 male flies of each genotype were divided into plastic vials in groups of 10, where each vial contained regular fly food. The flies were transferred into new vials with fresh fly food every 2–3 days and simultaneously, the number of dead and live flies was counted. This process was repeated until all flies had died. Kaplan–Meier survival curves [53] were generated using GraphPad Prism software 6 (GraphPad software Inc., San Diego, CA, USA) and longevity statistics were analyzed.

### 4.3. Dissection

The flies were decapitated and dissected in PBS solution under microscope using Jewelers forceps, Dumont No. 5 (Merck, Darmstadt, Germany). Both crop and proventriculus were still attached to the intestine after dissection for orientation purposes.

### 4.4. Antibody and Ligand Double Staining

The synthesis of ligand h-FTAA has been described elsewhere [33]. The intestine from Arctic-, T_22_Aβ1–42-, Aβ1–42 × 2-, and control flies was dissected as described above and placed in a well containing PBS on a glass microscope slide. The well was formed by the space between two coverslips that was attached on the slide prior to the staining procedure using nail polish. To avoid leakage, a hydrophobic pen was used to draw a water-repellant barrier in the top and bottom of the well, which was not enclosed by the cover slips. The intestines were fixed in 4% formaldehyde (Merck, Darmstadt, Germany) for 10 min at RT, washed in PBS (3 × 2 min, RT), and then incubated in PBS containing 0.1% triton x-100 (PBS-T) and 5% normal goat serum for 1 h at RT. The anti-human Aβ antibody (Mabtech, Nacka Strand, Sweden) was diluted 1:200 in PBS-T containing 5% normal goat serum and was added to the intestines. After 16 h of incubation at 4 °C, the samples were washed in PBS-T (3 × 10 min, RT) and then incubated for 1 h at RT with goat anti-mouse secondary antibody conjugated with Alexa 647 (Thermo Fisher Scientific Inc., Waltham, MA, USA) diluted 1:400 in the same buffer as the primary antibody. The intestines were washed in PBS (3 × 10 min, RT) and then stained with 3 μM h-FTAA, diluted in PBS, for 30 min at RT. After washing in PBS three times, the samples were incubated with 300 nM diamidino-2-phenylindole (DAPI, Ted Pella Inc., Redding, CA, USA) for 5 min at RT, washed with PBS three times, and then mounted using Dako mounting medium for fluorescence (Agilent, Santa Clara, CA, USA). The result was analyzed using an inverted Zeiss 780 laser scanning confocal microscope (Zeiss, Oberkochen, Germany). Prior to the analysis, the rims of the coverslip were sealed with nail polish.

### 4.5. Apoptotic Assay

The intestine from Arctic-, T_22_Aβ1–42-, Aβ1–42 × 2-, and control flies was dissected as described above and placed in a well containing PBS on a glass microscope slide (see Section 4.4). The samples were mounted using Dako mounting medium for fluorescence (Agilent, Santa Clara, CA, USA), and shortly after, analyzed using an inverted Zeiss 780 laser scanning confocal microscope (Zeiss, Oberkochen, Germany). Prior to the analysis, the rims of the coverslip were sealed with nail polish.

### 4.6. Statistical Analysis

The data were analyzed using GraphPad Software 9. Kaplan–Meier survival curves were generated using GraphPad Prism software 9.

## Figures and Tables

**Figure 1 ijms-25-02105-f001:**
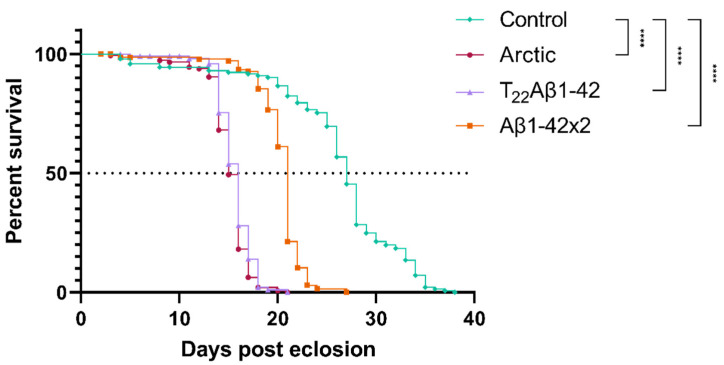
Longevity analyses showing toxic effects on Amyloid-β (Aβ)-expressing flies. Toxic effects were assessed by the longevity assay for Arctic- (red, circle; *n* = 146), T_22_Aβ1–42- (purple, triangle; *n* = 117), Aβ1–42 × 2- (orange, square; *n* = 143), and control flies (green, diamond; *n* = 145). The definition of significance was *p*-values of less than 0.0001 (****). Median survival times (50% survival, see dashed line) of Arctic-, T_22_Aβ1–42-, Aβ1–42 × 2-, and control flies were 15, 16, 21, and 27 days, respectively.

**Figure 2 ijms-25-02105-f002:**
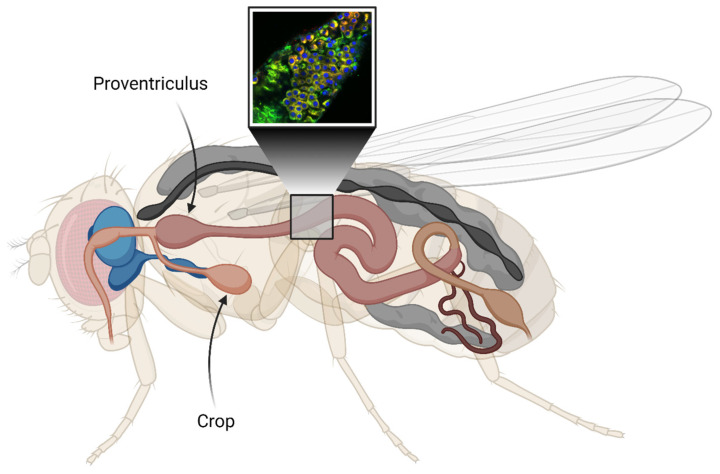
Illustration of the digestive tract (light pink) of *Drosophila melanogaster*. The cutout represents an estimation of the selected area used for the fluorescence image acquisition. The crop is an organ that resembles the mammalian stomach. Here, the central nervous system (CNS) is shown in blue. The cutout image shows co-staining of Aβ aggregates in the gut using an antibody against Aβ (red) and the amyloid-binding luminescent conjugated oligothiophene (LCO) ligand h-FTAA (green). DAPI (blue) is used to visualize cell nuclei. The figure is created with BioRender.

**Figure 3 ijms-25-02105-f003:**
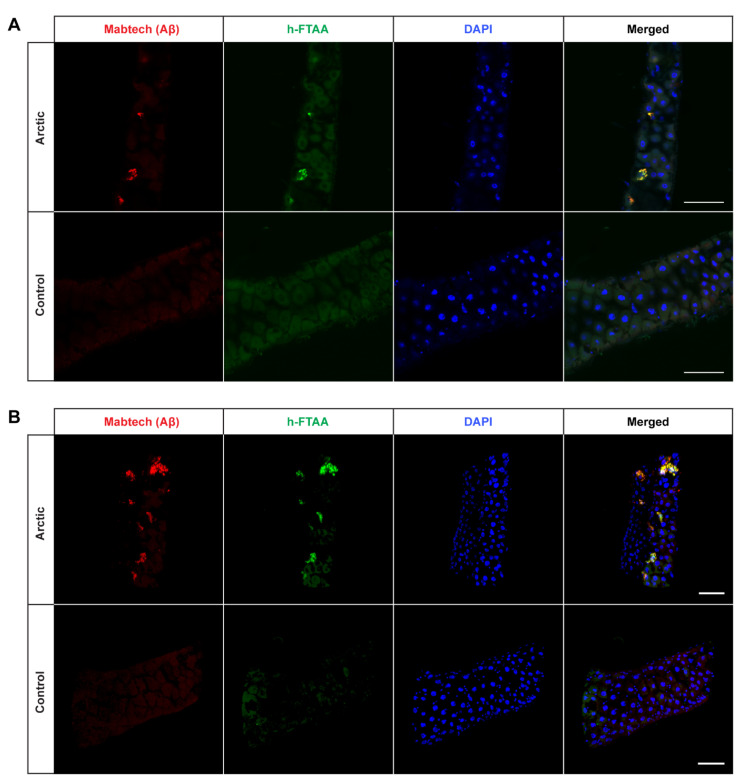
Detection of Aβ aggregates in the midgut of *Drosophila* flies expressing the Arctic mutant of the Aβ1–42 peptide in the enterocytes using the *Myo31DF* driver. (**A**) Confocal microscope single-plane images showing the midgut of Arctic- (top) and control flies (bottom) stained with Mabtech anti-human Aβ antibody (red) and LCO ligand h-FTAA (green) 16 days post eclosion. The sections have been counterstained with DAPI (blue) to visualize cell nuclei. Scale bar, 50 μm. (**B**) Confocal microscope 3D images of the same region as shown in (**A**). Scale bar, 50 μm.

**Figure 4 ijms-25-02105-f004:**
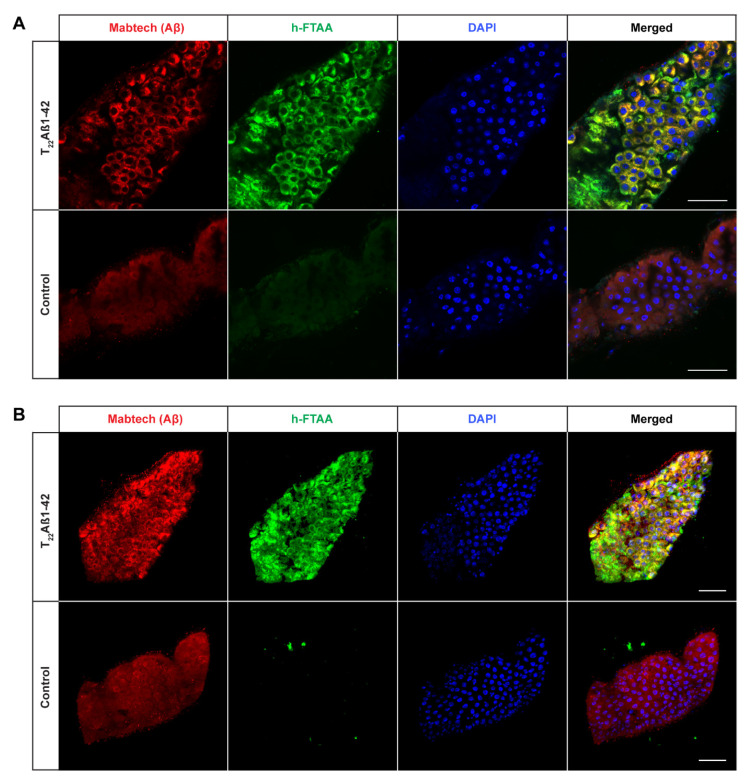
Detection of Aβ aggregates in the midgut of *Drosophila* flies expressing the tandem dimeric construct T_22_Aβ1–42 in the enterocytes using the *Myo31DF* driver. (**A**) Confocal microscope single-plane images showing the midgut of T_22_Aβ1–42- (top) and control flies (bottom) stained with Mabtech anti-human Aβ antibody (red) and LCO ligand h-FTAA (green) 15 days post eclosion. The sections have been counterstained with DAPI (blue) to visualize cell nuclei. Scale bar, 50 μm. (**B**) Confocal microscope 3D images of the same region as shown in (**A**). Scale bar, 50 μm.

**Figure 5 ijms-25-02105-f005:**
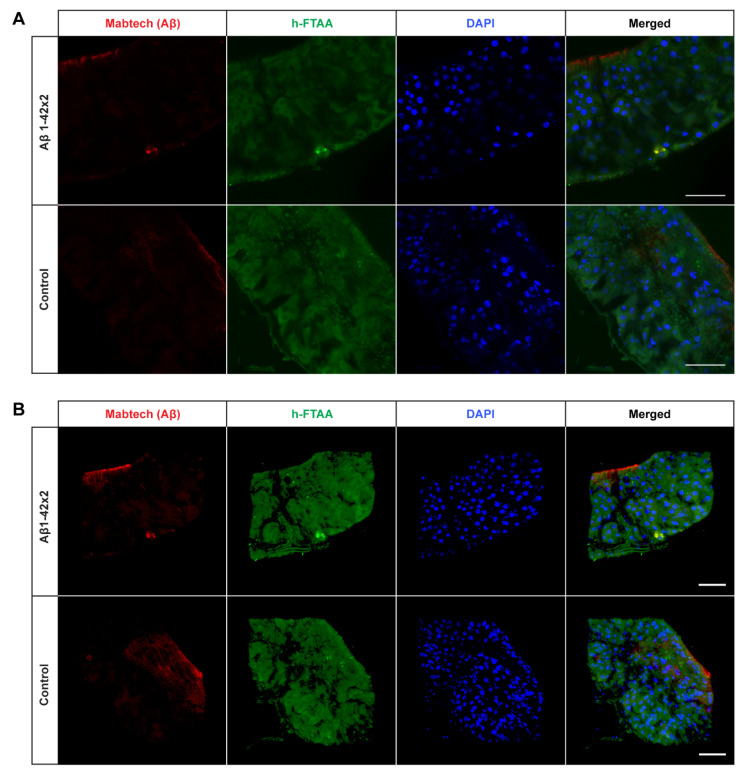
Detection of Aβ aggregates in the midgut of *Drosophila* flies expressing two copies of the Aβ1–42 peptide in the enterocytes using the *Myo31DF* driver. (**A**) Confocal microscope single-plane images showing the midgut of Aβ1–42 × 2- (top) and control flies (bottom) stained with Mabtech anti-human Aβ antibody (red) and LCO ligand h-FTAA (green) 21 days post eclosion. The sections have been counterstained with DAPI (blue) to visualize cell nuclei. Scale bar, 50 μm. (**B**) Confocal microscope 3D images of the same region as shown in (**A**). Scale bar, 50 μm.

**Figure 6 ijms-25-02105-f006:**
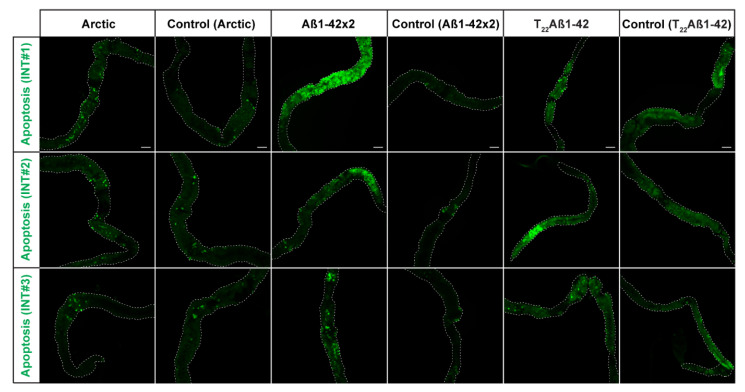
Activation of the apoptotic pathway in the midgut of *Drosophila* fly-expressing Aβ variants in the enterocytes using the *Myo31DF* driver: Arctic- (**left**), Aβ1–42 × 2- (**middle**) or T_22_Aβ1–42 (**right**) flies. In the fluorescence images, the GFP signal (green) is shown, which acts as a sensor of apoptosis. The analysis included three intestines (INTs) for each group, and it was performed at 15-, 21-, and 15-days post eclosion, respectively. The corresponding control flies are shown to the right of each genotype. The white, dotted line outlines the edges of the intestine. Scale bar, 100 μm.

## Data Availability

Data is contained within the article and Appendix A.

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
