# Peer review of "A Novel Drosophila Model of Alzheimer’s Disease to Study Aβ Proteotoxicity in the Digestive Tract"

_ijms, 2024, doi:10.3390/ijms25042105_

Round 1

Reviewer 1 Report

Comments and Suggestions for Authors

In this paper the authors report the development of a potential novel Drosophila AD model. Unlike the previously used AD models that were CNS specific, this model is based on Ab peptide expression in the gut cells. Similar to the classical AD models, expression of three different type of Ab peptides resulted in a proteotoxicity, reducing lifespan and causing protein aggregate deposition in the gut cells. Given that in the traditional drug screens when the candidates are orally administrated the blood brain barrier can prevent the full appreciation of the effect, I find this digestive tract based approach highly innovative and potentially valuable for the field. From formal aspects the paper is well written, the experiments are clearly described, the little statistics is just appropriate, and the figures are of sufficiently good quality. Despite that, the content of this paper is rather slim with a single set of survival test (with 4 genotypes only) and a couple of immunostainings. I think the paper would benefit a lot from a more through analysis of at least a few of the key findings.

My main questions and comments are the following:

1.         Number of the apoptotic cells caused by Ab peptide expression was analyzed only at one timepoint for each construct (Ab derivative). It would be useful to examine whether these numbers or patterns change with time.

2.         A surprising finding was that despite exhibiting the least strong toxicity, the Ab1-41x2 peptide caused the highest level of apoptosis. The authors speculated that it might be because the Arctic and the T22Ab1-41 constructs induced necrotic cell death instead of apoptosis. This might indeed be the case, however, I lack the efforts to test this hypothesis. First of all, the authors used only a single apoptosis marker, it would be important to confirm this key finding of the paper with at least one more marker. In addition, the authors should use other markers to distinguish between apoptosis versus necrosis. Such markers do exist, even if they may not be so perfect in flies as in mammalian cells.

3.         An additional way to address the above issue would be to try to rescue the apoptotic and other defects observed upon Ab expression by UAS-p35 (the caspase inhibitor) expression.

4.         The analysis of gut cell morphology is yet another marker that could give hints as to the unexpected contradiction as to the effect of Ab1-41x2 (toxicity versus apoptosis).

5.         As a proof of principle experiment it should be tested whether the previously reported Congo Red and/or Curcumin are able to reduce the toxicity/number of aggregates in this gut specific AD model as well. It would be an important experiment to assess the real potential of this gut based system as a platform for small molecule screens.

Author Response

The comments have been addressed and are summarized in the attached file.

 Best regards

Reviewer 2 Report

Comments and Suggestions for Authors

In this manuscript, authors designed a novel Drosophila model that express the Aβ peptides in the digestive tract. They also developed a built-in sensor which provides a fluorescent signal from the green fluorescent protein as an indicator of apoptotic activity.

Title: I suggest removing the word designing from the title.

Results

-       Figures are not inserted in the text, please insert each in the appropriate place in the text. Error! Reference source not found. Please correct this.

-       Line 118: 16, 21 not 16 21

Conclusions

-       What does this sentence mean? “since expression of Aβ in the digestive tract increases the possibility that the drugs or substances reach the target site”. Please restate it for clarity. Expression of Aβ in the digestive tract increases the possibility that the drugs interact with Aβ, this does not mean that the drug can reach the target site. Drug delivery to target site is different from drug ability to interact with prion.

Methods

-       Please add statistical analysis subsection at the end of methods section which indicates the statistical tests used in the study.

Comments on the Quality of English Language

Minor English editing.

Author Response

(The authors gave the same response as above.)

Round 2

Reviewer 1 Report

Comments and Suggestions for Authors

Although not experimentally, but my concerns were addressed by the authors, and I support the publication of this study.